# Morphology, Mechanical, and Water Barrier Properties of Carboxymethyl Rice Starch Films: Sodium Hydroxide Effect

**DOI:** 10.3390/molecules27020331

**Published:** 2022-01-06

**Authors:** Pornchai Rachtanapun, Sarinthip Thanakkasaranee, Rafael A. Auras, Nareekan Chaiwong, Kittisak Jantanasakulwong, Pensak Jantrawut, Yuthana Phimolsiripol, Phisit Seesuriyachan, Noppol Leksawasdi, Thanongsak Chaiyaso, Sarana Rose Somman, Warintorn Ruksiriwanich, Warinporn Klunklin, Alissara Reungsang, Thi Minh Phuong Ngo

**Affiliations:** 1School of Agro-Industry, Faculty of Agro-Industry, Chiang Mai University, Mae-Hea, Mueang, Chiang Mai 50100, Thailand; sarinthip.t@cmu.ac.th (S.T.); meen.nareekan@gmail.com (N.C.); jantanasakulwong.k@gmail.com (K.J.); yuthana.p@cmu.ac.th (Y.P.); phisit.s@cmu.ac.th (P.S.); noppol@hotmail.com (N.L.); thachaiyaso@hotmail.com (T.C.); warinporn.k@cmu.ac.th (W.K.); 2Cluster of Agro Bio-Circular-Green Industry (Agro BCG), Chiang Mai University, Chiang Mai 50100, Thailand; pensak.amuamu@gmail.com (P.J.); Sarana.s@cmu.ac.th (S.R.S.); warintorn.ruksiri@cmu.ac.th (W.R.); 3Center of Excellence in Materials Science and Technology, Faculty of Science, Chiang Mai University, Chiang Mai 50200, Thailand; 4School of Packaging, Michigan State University, East Lansing, MI 48824, USA; aurasraf@msu.edu; 5Materials Science Research Center, Faculty of Science, Chiang Mai University, Chiang Mai 50200, Thailand; 6Department of Pharmaceutical Sciences, Faculty of Pharmacy, Chiang Mai University, Chiang Mai 50200, Thailand; 7Plant Bioactive Compound Laboratory (BAC), Department of Plant and Soil Sciences, Faculty of Agriculture, Chiang Mai University, Chiang Mai 50200, Thailand; 8Department of Biotechnology, Faculty of Technology, Khon Kaen University, Khon Kaen 40002, Thailand; alissara@kku.ac.th; 9Research Group for Development of Microbial Hydrogen Production Process, Khon Kaen University, Khon Kaen 40002, Thailand; 10Academy of Science, Royal Society of Thailand, Bangkok 10300, Thailand; 11Department of Chemical Technology and Environment, The University of Danang-University of Technology and Education, Danang 550000, Vietnam; ntmphuong@ute.udn.vn

**Keywords:** mechanical properties, morphology, water vapor permeability, solubility, NaOH

## Abstract

Carboxymethyl rice starch films were prepared from carboxymethyl rice starch (CMSr) treated with sodium hydroxide (NaOH) at 10–50% *w*/*v*. The objective of this research was to determine the effect of NaOH concentrations on morphology, mechanical properties, and water barrier properties of the CMSr films. The degree of substitution (DS) and morphology of native rice starch and CMSr powders were examined. Fourier transform infrared spectroscopy (FT-IR), X-ray diffraction (XRD), and differential scanning calorimetry (DSC) were used to investigate the chemical structure, crystallinity, and thermal properties of the CMSr films. As the NaOH concentrations increased, the DS of CMSr powders increased, which affected the morphology of CMSr powders; a polyhedral shape of the native rice starch was deformed. In addition, the increase in NaOH concentrations of the synthesis of CMSr resulted in an increase in water solubility, elongation at break, and water vapor permeability (WVP) of CMSr films. On the other hand, the water contact angle, melting temperature, and the tensile strength of the CMSr films decreased with increasing NaOH concentrations. However, the tensile strength of the CMSr films was relatively low. Therefore, such a property needs to be improved and the application of the developed films should be investigated in the future work.

## 1. Introduction

Carboxymethyl starch (CMS) was firstly modified in 1924 by the reaction of starch with sodium monochloroacetate in an alcohol solution [1,2]. In general, water solubility increased as the substitution level (DS) increased. CMS is widely used in food, cosmetic, and pharmaceutical industries. Among starch derivatives, CMS is of particular interest because of its outstanding properties. For example, CMS is commonly used as a thickener in the formulation of textile printing pastes. In addition, CMS is typically used as an additive in the paper industry and a water-soluble polysaccharide [3]. The DS is the average number of functional groups introduced into the anhydroglucose unit (AGU) and could be produced with a very wide variety of carboxymethyl group substitutions [4]. CMS synthesized from potato starch had excellent solubility in cold water and high viscosity due to the high DS value [5]. Therefore, etherified starch derivatives received great attention [6]. CMS is mostly used as a stabilizing agent in food production applications such as ice cream, vegetables, and drinks. Furthermore, CMS is a preservative in fresh meat products and crops. CMS is applied as an additive in non-food manufacturing (i.e., resizing and printing in the cloth industry [7,8] and controlled drug release suspending agent in the pharmaceutical industry [9]). CMS is also used as a binder and tablet film for covering medicines, medicine formulation, and gel-based coating materials [10,11]. In addition, CMS was introduced in polymers to obtain a hydrophilic behavior of film, i.e., carboxymethyl cellulose [12,13], carboxymethyl chitosan [14], and carboxymethyl starch [15,16]. Moreover, CMS-based films are soluble in cold water and their physicochemical properties are dependent on their DS values [3].

Rice starch is a highly intensive starting material for producing edible and biodegradable films because it is a natural polymer and mass production is possible from sustainable agricultural resources [17]. Moreover, rice starch is renewable, low cost, and it can partially or fully replace other edible and biodegradable polymers [18]. Although, the function of native rice film is specified because the film is brittle, slightly opaque, and does not dissolve in cold water [19]. To overcome such problems, plasticizers are added during the film creation process to increase flexibility and reduce internal hydrogen bonds between polymer chains, which increase free volume and the gap between molecular chains of polymer [20]. The most generally used plasticizers in starch-based films are polyols; for example, glycerol, poly(ethylene glycol), and sorbitol, which are mostly used due to their hydrophilic properties [21]. Glycerol-plasticized films show higher solubility [22], moisture absorption, and flexibility [23] than sorbitol-plasticized films. Laohakunjit et al. [24] reported that rice starch films plasticized with sorbitol had better oxygen barrier properties than films plasticized with glycerol. Additionally, rice starch was also combined with alternative materials; for example, agar [25], gelatin [26], chitosan [27,28], methylcellulose [29], carboxymethyl cellulose [12], and carboxymethyl chitosan [28,30]. Moreover, starch modification under chemical reaction is one of the main methods which has been employed to develop starch films with unique characteristics and properties [31]. The addition of chitosan into rice starch increased the water barrier property, but no changes in the mechanical properties of rice starch–chitosan films [27]. Introduction of carboxymethyl chitosan into rice starch improved the mechanical properties and thermal stability of films [28]. The incorporation of propolis extract into rice starch/carboxymethyl chitosan films enhanced their antioxidant and antimicrobial properties [30]. Moreover, Rachtanapun et al. [31] studied the effect of sodium hydroxide (NaOH) concentrations on properties of carboxymethyl rice starch (CMSr). They found that the morphology of the CMSr granules was deformed when the NaOH concentration increased, which was correlated with DS. In addition, viscosity increased, whereas the crystallinity of the CMSr decreased. The effect of reaction parameters, amount of NaOH, sodium monochloroacetate, and reaction time on the DS of carboxymethyl yam (*Dioscorea esculenta*) starch was studied by Nattapulwat et al. [32]. The experimental results showed that the optimal ratio of NaOH and sodium monochloroacetate to anhydroglucose unit was 1.80 and 2.35, respectively, for 4.8 h, which succeeds the DS for 0.19.

Although, a few studies have been carried out regarding the effect of NaOH concentrations on the properties of CMSr powders, we attempted to investigate the effect of NaOH concentrations on water solubility, functional group, morphology, thermal analysis, contact angle, mechanical properties (elongation at break and tensile strength), and water vapor permeability (WVP) of CMSr films.

## 2. Results and Discussion

### 2.1. Degree of Substitution (DS) of CMSr Powder

The effect on different NaOH concentrations on the DS of the CMSr powder was examined. The result found that the DS value of CMSr powder increased with higher NaOH levels. The value of DS showed 0.08, 0.23, 0.31, 0.38, and 0.32 with 10%, 20%, 30%, 40%, and 50% NaOH concentrations, respectively. However, with the 50% NaOH concentration, the DS of CMSr powder was gradually decreased. The DS value could be explained by a two-step reaction of carboxylation. The first step is the alkalization of starch, where sodium hydroxide reacts with the hydroxyl group of rice starch molecules and is converted to aloxides [31]. The second step is generated from the first step at the strongest alkaline concentration. Etherification also occurs as a side reaction; the reaction between sodium hydroxide and sodium monochloroacetate to sodium glycolate form [1]. In general, an increase in DS because of a reduction of crystallinity in the polysaccharides facilitates the carboxymethylation reaction affected by a high concentration of base and acid [6]. Conversely, the DS decreases because of the side reactions dominated by high sodium glycolate by-products [33,34]. An increase in DS affects the water solubility in polysaccharides, such as cellulose [13,15,35], chitosan [14,36], and starch [16,31]. Moreover, the DS of carboxymethylation depends on particle sizes [14], solvents [37], types of polymers [36,38], and times and temperatures of reactions [39]. Nattapulwat et al. [32] studied the effect of NaOH, sodium monochloroacetate, and reaction time on DS of carboxymethyl yam starch (*Dioscorea esculenta*). The optimal ratio of NaOH (1.80) to sodium monochloroacetate (2.35) was used for 4.8 h, which resulted in a DS of 0.19. There are also previous studies relating to carboxymethyl cellulose that determined the effect of NaOH concentrations on the degree of substitution (DS). For example, carboxymethyl cellulose from asparagus stalk end (CMCas) with various NaOH concentrations by Klunklin et al. [33]. The results showed that CMCas at a concentration of 30% of NaOH for the carboxymethylation reaction had a maximum DS of 0.98. In addition, Rachtanapun et al. [34] studied carboxymethyl cellulose from nata de coco (CMCn) with *Acetobacter xylinum* starting NaOH concentrations from 20% to 60%. In the carboxymethylation process, optimal conditions include using a NaOH content of 30 g/100 mL as the highest DS value (0.92). Including the experiment of Rachtanapun et al. [13], carboxymethyl durian rind (CMCd) was synthesized using different concentrations of NaOH. The results showed that the DS value of CMCd increased with increasing NaOH concentration and obtained a maximum DS of 0.87 at a 30% (*w*/*v*) NaOH concentration.

### 2.2. Scanning Electron Microscopy of CMSr Powders

The morphological characteristic of the CMSr powders with different NaOH concentrations was examined using SEM. As shown in Figure 1, native rice starch showed several individual granules with a polyhedral form and smooth surface (Figure 1a,A). The size and appearance of the granules began to change when the NaOH concentration was increased. The granules of CMSr treated with NaOH concentrations of 10% and 20% were similar to the native rice starch with slight damage as shown in Figure 1b,B,c,C, but at 20% NaOH concentration, the granules started agglomeration (Figure 1c). Conversely, the CMSr powder treated with the NaOH concentration (30% and 40%) showed individual granules with irregular shapes (Figure 1d,D,e,E). At 50% NaOH concentration, the CMSr powder exhibited an agglomerated form (Figure 1f,F), whereas the CMSr treated with 60% NaOH showed a gel-like aspect in which isolated granules of CMSr obviously disappeared (Figure 1g,G). This result is the same as that found in carboxymethyl cassava starch [40]. As the NaOH concentration increased, the damage of the surface area of the rice starch granules increased because alkaline solutions may reduce the strength and stability of the granular molecular arrangement, resulting in a loss of granulation [31]. Alkalization changed the granules, making them weaker with a loss of crystallinity, and thereby allowing the etherifying agents higher entrance to the starch molecules in carboxymethylation processes [6]. Therefore, this result corresponds to the DS value, leading to a correlation between the NaOH concentration with the DS of the CMSr [31].

### 2.3. FT-IR of Native Rice Starch Film and CMSr Films

FT-IR was used to analyze functional group variations in the native rice starch film and CMSr films, as shown in Figure 2. The substitution reaction of CMSr via the carboxymethylation is related to change in functional groups, including the hydroxyl group (–OH stretching), the C–H stretching carbonyl group (C=O stretching), hydrocarbon groups (–CH_2_ scissoring), and ether groups (–O– stretching) at 3200–3600, 3000, 1600, 1450, and 1000–1200 cm^−1^, respectively [31,33,34]. The absorption bands of CMSr treated with 10% NaOH was similar to native rice starch film. The intensity of the carbonyl group (C=O stretching) and hydrocarbon groups (–CH_2_ scissoring) of CMSr films slightly increased with increasing NaOH concentrations ranging from 10% to 20%, whereas such functional groups remarkably increased at the higher NaOH concentrations (30–50%). These significant changes confirmed that carboxymethylation took on the rice starch molecules, which was similar to the carboxymethylation of mung bean starch [11] and yam starch [32]. However, the CMSr powder treated with 60% NaOH could not form as a film due to the side reaction effect between NaOH and sodium monochloroacetate forming sodium glycolate, which is consistent with the result of Lawal et al. [6]. Thus, the properties of sample treated with 60% NaOH was not investigated.

### 2.4. SEM of Rice Starch Film and CMSr Films

Scanning electron micrographs of cross-sections of rice starch film and CMSr films with different NaOH concentrations are shown in Figure 3. The film was cut with liquid nitrogen to obtain a cross-section. The rice starch film showed a rough surface and a small granule, which may be caused by a low water solubility of rice starch powder in film forming [41], as in Figure 3a,A. The surface of the CMSr film treated with a 10% NaOH concentration presented an uneven surface (Figure 3b,B) due to the poor water solubility of CMSr powder affected by a lower DS (0.08). This is consistent with the morphology of CMSr powder with remaining crystalline granules (Figure 1b,B), which could not completely dissolve in water during the film forming. Conversely, an increase in the NaOH concentration (e.g., 20–50% NaOH) apparently affected the morphology of the CMSr films, resulting in a smoother film surface (Figure 3c–f,C–F) due to a higher DS (in the range of 0.23–0.38 for 20–50% NaOH). This is related to destruction of the crystalline structure of the starch granules and the formation of carboxymethyl groups on their surface as described in SEM micrographs (Figure 1c–f,C–F) and FT-IR spectra [15,34], leading to their easy dissolving in water during film forming. However, the CMSr powder treated with 60% NaOH could not form as a film because of the consequent collapse of the crystal structure affected by the side reaction due to the excess of NaOH as explained in the previous section.

### 2.5. X-ray Diffraction (XRD) of Native Rice Starch Film and CMSr Films

XRD patterns of the native rice starch film and the CMSr films with different NaOH concentrations are shown in Figure 4. The diffraction pattern of native rice starch films showed characteristic peaks at 14.9, 17.0, 18.0, and 22.8° 2θ, corresponding to a C-type crystalline pattern [31,42]. The CMSr films with a 10% NaOH concentration showed a similar pattern compared to the native rice starch film, but lower intensity. At higher NaOH concentrations (20–50% NaOH), the C-type crystalline pattern [31,42] of the CMSr films disappeared due to loss of crystallinity, attributing to the rupture of starch granules during the modification through carboxymethylation [33], as shown in SEM results (Figure 1). In general, NaOH treatment decreases the crystallinity of polysaccharide polymers caused by the breaking of hydrogen bonds [34]. The present research is also consistent with other works, such as carboxymethyl cassava starch [15], carboxymethyl rice starch [31], carboxymethyl cellulose from asparagus stalk ends [33], and carboxymethyl cellulose from nata de coco [34].

### 2.6. Differential Scanning Calorimetry (DSC) of Native Rice Starch Film and CMSr Films

In general, crystallinity plays a significant role in the material’s barrier and mechanical properties, which can be investigated by DSC [43]. The thermal property of native rice starch and CMSr films with different NaOH concentrations is shown in Figure 5. The melting temperature (T_m_) of native rice starch film was 173.7 °C. The T_m_ of CMSr films treated with 10% and 20% NaOH were 168.2 and 154.7 °C, which were lower than that of the native rice starch film. The T_m_ of the CMSr films synthesized with 30%, 40%, and 50% NaOH were 167.0, 135.6, and 108.6 °C, respectively. The area under the endothermic peak expresses the heat (enthalpy) of fusion (ΔH), reflecting the crystallinity of the polymeric films [44]. The native rice starch film showed a sharp endothermic peak and a high heat of fusion. The endothermic peak of CMSr films treated with lower NaOH concentrations (10–20% NaOH) were similar to that of the native rice starch film but shifted toward low temperatures together with the reduction of ΔH, indicating a reduction of crystallinity of films. This phenomenon occurred because the intermolecular force (i.e., H-bond) between rice starch molecules was slightly disturbed from the formation of bulky groups (i.e., carboxymethyl group), affecting the chain arrangement of the native starch and number of crystalline granules. At higher NaOH concentrations (30–50% NaOH), the endothermic peak of CMSr films became broader and remarkably shifted toward low temperatures, implying the loss of rice starch’s crystallinity, attributed to a great substitution of the carboxymethyl group. As previously described, the higher NaOH concentrations strongly affected the intermolecular force by breaking the H-bonding between rice starch molecules, resulting in the disruption of the crystalline structure of the native rice starch molecules [45]. This phenomenon facilitated the carboxymethylation reaction between sodium monochloroacetate and rice starch molecules, leading to the greater substitution of the carboxymethyl group on the rice starch molecules [46]. This result was consistent with the DS, FT-IR, XRD, and SEM results. In addition, the present result was also similar with carboxymethyl cellulose from asparagus stalk ends [33], carboxymethyl bacterial cellulose from nata de coco [34], and carboxymethyl cellulose powder and films from *Mimosa pigra* peel [35].

### 2.7. Percentage of Soluble Matter (%SM) of Native Rice Starch Film and CMSr Films

Figure 6 shows the effect of NaOH concentrations on the %SM of CMSr films. The native rice starch film has a very low %SM (2.07), which indicated an insolubility in water. At lower NaOH concentrations (10% and 20% NaOH), the %SM of CMSr films slightly increased (3.58 and 3.82), but were mostly insoluble in water, indicating a lower formation of the carboxymethyl group. Obviously, the CMSr films treated with higher NaOH concentrations (30–50% NaOH) exhibited higher %SM (89.49, 95.72, and 99.70 for 30%, 40%, and 50% NaOH treatments, respectively), suggesting an increase in water solubility of the CMSr films. This implied that the %SM of the CMSr films was dependent on the DS and morphological structure of the synthesized CMSr powders affected by higher NaOH concentrations, as explained in the DS, FT-IR, and SEM results. This confirmed that the sufficient NaOH concentration resulted in a higher carboxymethylation reaction and a greater polarity of the CMSr films [14]. The finding is similar to the investigation of sodium carboxymethyl mung bean starch granules [9].

### 2.8. Contact Angle of Native Rice Starch Film and CMSr Films

The water contact angle is the most common parameter used to describe the hydrophilicity of film surfaces [47]. Hydrophilicity of polymeric film can be investigated by the water contact angle. As shown in Figure 7 and Table 1, a dynamic water contact angle of all films decreased with time (0–50 s) due to the chemical affinity between water and polymeric films [48]. The dynamic water contact angles of the native rice starch were slightly changed in the range of 93.9–90.9 with time. The dynamic water contact angles of the CMSr films treated with 10–20% NaOH concentrations were similar to that of the native rice starch, but relatively lower water contact angles. At higher NaOH concentrations (30–50% NaOH), the water contact angle of CMSr films notably decreased with time, and lower than that of the native rice starch and those treated with lower NaOH concentrations. The lower dynamic water contact angle implied the enhancement of the hydrophilic behavior of the CMSr films treated with higher NaOH concentrations, attributed to the higher formation of the carboxymethyl group [2,47,49]. This is consistent with the results of DS, FT-IR, and the soluble matter (%SM) of CMSr films.

### 2.9. Water Vapor Permeability (WVP)

The effect of different NaOH concentrations on the WVP of CMSr films is shown in Figure 8. Notably, the WVP of the CMSr films was significantly dependent on NaOH concentrations used in the carboxymethylation reaction. As the NaOH concentrations increased (10–30% NaOH), the WVP of the CMSr films slightly increased. This indicated a slight increase in hydrophilicity of the CMSr films due to a little formation of the carboxymethyl group (polar group) on the rice starch molecules and a reduction of crystallinity in the CMSr films [3,28,50]. Whereas at very high concentrations (40–50% NaOH), the WVP of the CMSr films was obviously increased, suggesting a significant improvement of hydrophilicity character due to the greater formation of the carboxymethyl group on the starch molecules and the loss of crystallinity in the CMSr films [2,6,29]. This result is in agreement with the DS, FT-IR, and dynamic water contract angle results, which confirmed the enhancement of polarity and hydrophilicity of the CMSr films.

### 2.10. Tensile Strength (TS) and Elongation at Break (%E)

The TS and %E of polymeric films are commonly investigated, in which the mechanical property is dependent on the crystallinity, intermolecular forces, and ionic character [33,51,52]. The different NaOH concentrations affected the TS values and %E of CMSr films as shown in Table 2. The native rice starch exhibited the highest TS, but lowest %E, indicating the behavior of the brittle polymeric film due to the strong intermolecular force (H-bond) between the native rice starch molecules and the high crystallinity in the native rice starch film as described in the XRD and DSC results. At lower concentrations (20–50%), the TS of the CMSr films slightly decreased due to the reduction of crystallinity, relating to the destruction of crystalline granules of the native rice starch [30,32,41]. However, as the NaOH concentration (30–50%) increased, the TS of the CMSr notably decreased because of the reduction of intermolecular force between rice starch molecules as well as the reduction and loss of crystallinity, as explained in the XRD and DSC results.

The %E of the CMSr films slightly increased with increasing NaOH concentrations, indicating the behavior of flexible CMSr films. This was attributed to reduction of crystallinity. Nawaz et al. [53] reported several factors that affect the chemical properties, physical properties, and applications of starch. It has been found to induce changes in the crystalline starch amorphous form in starch gel formation. The amount of water in the CMSr films increased, increasing the %E. These reasons agree with the observations in the work of Phan et al. [25]. Thus, CMSr films led to decreased TS, but they increased %E with increasing NaOH concentrations.

Moreover, there is research that describes the relationship between DS values for TS and %E. Phan et al. [25] reported that the TS is presented to the rise of the DS since replacing the methyl group, carboxymethyl, makes an expansion in the ionic type and intermolecular forces between the polymer groups. However, Klunklin et al. [33] explained that with a greater concentration of NaOH, TS was lowered because of sodium glycolic acid, a by-product from the reaction of CMC and synthetic biodegradable polymer. Rachtanapun et al. [35] indicated that the relation of CMC films from *Mimosa pigra* showed that the TS of CMC films from *Mimosa pigra* increased with increased NaOH concentrations (20–30 g/100 mL) and Rachtanapun et al. [54] studied CMC film from mulberry paper waste and found that with the concentration of NaOH increasing, TS increased, and may cause hydrolysis of the cellulose chain. 

## 3. Materials and Methods

### 3.1. Materials

Native rice starch was obtained from the Thai Flour Industry Co., Ltd. (Bangkok, Thailand). Analytical grade glacial acetic acid, hydrochloric acid, isopropanol, sodium chloride, sodium hydroxide (Merck KGaA, Darmstadt, Germany), and chloroacetic acid (Sigma-Aldrich, Darmstadt, Germany) were used as received, and all were of commercial grade. Methanol, absolute, 99.8%, reagent was purchased from the Northern Chemical Co., Ltd. (Chiang Mai, Thailand).

### 3.2. Synthesis of Carboxymethyl Rice Starch (CMSr)

The synthesis of CMSr was conducted by following the method detailed in Rachtanapun et al. [31]. Briefly, 400 mL of isopropyl alcohol was dissolved in 30 g of monochloroacetic acid. Then, it was added to 100 g of native rice starch and stirred well. Different concentrations of NaOH (10%, 20%, 30%, 40%, 50%, and 60% *w*/*v* of water) were studied, then heated for 20 min at 50 °C. Glacial acetic acid was added to neutralize the solution. Finally, the solution was filtered and rinsed 4 times with 95% methanol in the CMSr purification. The modified CMSr was evaporated at 50 °C for 17 h and left through an 80-mesh sieve (Figure 1).

### 3.3. Degree of Substitution (DS)

The CMSr’s DS was performed following Klunklin et al. [33] and Rachtanapun et al. [34] by the USP XXIII method. Calculation of the number of hydroxyl groups replaced by carboxymethyl groups and sodium carboxymethyl groups at C2, 3, and 6 in the cellulose structure was used. The equation for finding the DS value is as follows (1):(1)DS=A+S 
where *A* is the *DS* of carboxymethyl acid and *S* is the *DS* of sodium carboxymethyl. *M* is consumption of the titration to end point (mEq) and *C* is the number of ash after ignition (%) as shown in Equations (2) and (3).
(2)$$A=\frac{1150M\;content}{\left(7120-412M-80C\right)\;content}$$
(3)$$S=\frac{\left(162+58A\right)C\;content}{\left(7120-80C\right)\;content}$$

### 3.4. Preparation of Native Rice Film and CMSr Films

In this study, 1.5 g of native rice and CMSr powders with different NaOH concentrations (3% *w*/*v*) was individually dissolved in 50 mL distilled water, and was stirred under 70 °C for 10 min. After that, sorbitol (30% *w*/*w* of powders) was added into the solution and was continuously stirred. Then, the resultant solution was degassed to release all air bubbles using an ultrasonic bath (the Elmasonic S series model S10H, J.J. Science Lab Co., Ltd., Bangkok, Thailand). The solution was cast in flat plastic plates (15 × 15 cm) to produce the films, following drying overnight at 50 °C. Then, the CMSr films were removed from the plates. The CMSr films with 12–14% of moisture content were kept at 27 ± 2 °C and 54 ± 2% relative humidity (RH) for 24 h [31].

### 3.5. Characterizations

#### 3.5.1. Fourier Transform Infrared Spectroscopy (FT-IR)

The FT-IR spectra of CMSr films with different NaOH concentrations were recorded at room temperature using a Nicolet 6700 FT-IR spectrometer (Thermo Electron Corporation, Waltham, MA, USA) in the range of 4000–400 cm^−1^ with 64 scans. The DTGS KBr detector and KBr beam splitter were used to investigate the functional groups of the CMSr film [55].

#### 3.5.2. Scanning Electron Microscopy (SEM)

An LV-scanning electron microscope (JSM 5910 LV, JEOL Ltd., Tokyo, Japan) [13] was used to investigate the surface morphology of rice starch and CMSr powder, as well as the cross-section microstructure of CMSr films. Prior to analysis at an acceleration voltage of 15 kV, the powders were coated with gold, whereas the films were frozen in liquid nitrogen, fractured, and then coated with gold.

#### 3.5.3. X-ray Diffraction (XRD)

The XRD pattern and crystallinity of the CMSr films with different NaOH concentrations were recorded in the reflection mode on an X-ray diffractometer (MiniFlex II, Rigaku, Tokyo, Japan). The scattering angle (2θ) was scanned from 5 to 60° at a scan rate of 5°/min [33].

#### 3.5.4. Differential Scanning Calorimetry (DSC)

The DSC thermogram of the films was determined using a DSC Q100 (TA Instruments, New Castle, DE, USA). Prior to analysis, films were conditioned at 23 ± 2 °C and 50 ± 10% RH for 48 h. Five milligrams of sample was used and heated from room temperature to 200 °C at a heating rate of 10 °C/min, adapted according to the method of Thanakkasaranee et al. [14]. Nitrogen was operated as a purge gas with a flow rate of 50 mL/min. The report was repeated at least 3 times.

#### 3.5.5. Film Solubility

A method was modified from Phan et al. [25]. Native rice starch and CMSr film was dried at 105 °C for 24 h and then kept in a desiccator. The sample was weighed close to 0.2000 g to determine the initial dry weight (W_i_). Using 50 mL of distilled water, the film sample was dissolved and shaken at 500 rpm for 15 min. Each film was suspended in 50 mL of distilled water and shaken at 500 rpm for 15 min. Then, the film solution was poured onto a weighted filter paper (Whatman, No. 93). The film solution was dried at 105 °C in the oven for 24 h and weighed again to obtain the final dry weight (W_f_). The following equation was used to calculate %SM of the films (Equation (4)):(4)$$\%\mathrm{SM}=\frac{\left(W_{i}-W_{f}\right)}{W_{i}}\times 100$$

#### 3.5.6. Dynamic Water Contact Angle

The dynamic water contact angle of the films was determined using a drop shape analysis (DSA30E, Krüss Co. Ltd., Hamburg, Germany) at 25 °C. The water droplet with a volume of 10.0 ± 0.5 µL was dropped on a solid surface and an image was taken every 10 s for 50 s [14].

#### 3.5.7. Film Thickness

The film thickness was evaluated with micrometers (model GT-313-A, Gotech Testing Machines Inc., Taichung City, Taiwan). The CMSr film prepared after NaOH treatment was compared to a CMSr film plasticized with sorbitol. Each sample was measured in five different areas and the average thickness was used for the tensile strength, %elongation at break, and water vapor permeability.

#### 3.5.8. Water Vapor Permeability

The CMS_r_ films with sorbitol as plasticizer were calculated at 25 °C using the method described by Klunklin et al. [33]. The test film was sealed into aluminum cups, each having an 8 cm diameter and 2 cm depth. A circular film sample (7 cm diameter) was taken, each cup was covered with 10 g of dry silica gel, and paraffin wax was used to close the cup. The sealed cups were stored in the desiccator at 52 ± 2% RH at 23.6 °C. Any differences in the sealed cup’s weight were reported every 24 h for 7 days and with a precision of 0.001 g. Weight gain to the cups over time (slope) was estimated. The following equation was applied to calculate the *WVTR* with a film area value at 28.27 cm^2^ (Equation (5)):(5)$$WVTR=\frac{slope}{film\;area}$$

The *WVP* (g m^2^ × 10^−4^ m^−2^ day^−1^.mmHg) was calculated using Equation (6):(6)$$WVP=\frac{WVTR\times L}{\Delta P}$$
where *L* is the film thickness mean (mm), and Δ*P* is the partial difference in steam pressure (mmHg) by measuring on both sides of the film sample (the vapor pressure of pure water at 23.6 °C = 21.7782 mmHg). Samples were analyzed in triplicate.

#### 3.5.9. Tensile Strength

Tensile strength (TS) and percentage elongation at break (%E) was measured following the method of Rachtanapun et al. [34] using a Universal Testing Machine Model 1000 (H1K-S, London, UK). The film sample was cut into strips of size 10 × 100 mm. All the film strips were equilibrated at 52 ± 2% RH for 2 days at 25 ± 2 °C. The means of 10 replicates of the test were reported.

### 3.6. Statistical Analysis

The effect of NaOH concentrations on the properties of the CMSr film were investigated using SPSS software version 20.0 0 (SPSS Inc., Chicago, IL, USA). All measurements were analyzed in triplicate. The data were presented as the mean ± SD. Analysis of variance (ANOVA) was used to show a significant difference (*p* ≤ 0.05) by Duncan’s multiple range test (DMRT).

## 4. Conclusions

The different NaOH concentrations used in the chemical modification of the native rice starch powder affected the morphology and DS of the CMSr powders, as well as the morphology, mechanical, and water barrier properties of the CMSr films. The increase in NaOH concentration (10–40%) resulted in an increase in DS of the CMSr powders. However, at 50% NaOH concentration, the DS decreased due to the partial side reactions by sodium glycolate (by-products). The higher NaOH concentrations (30–60%) obviously changed the morphology of the CMSr powders. At 60% NaOH concentration, the CMSr film could not be formed due to the side reaction effect between NaOH and sodium monochloroacetate forming sodium glycolate. The increase in NaOH concentrations resulted in the increase in %SM and WVP, but lower dynamic water contact angle of the CMSr films. The CMSr film treated with lower NaOH concentrations (10–20%) exhibited the hydrophilic character similar to that of the native rice starch film. In addition, the Tm and crystallinity of the CMSr films decreased, which resulted in the decrease in TS and increase in %E. This study indicated that the CMSr films became more hydrophilic and flexible through the carboxymethylation reaction, in which such properties of the CMSr films can be controlled by NaOH concentrations during the chemical modification of the native rice starch powder.

## Data Availability

The data presented in this study are available on request from the corresponding author.

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
