# Peer review of "Morphology, Mechanical, and Water Barrier Properties of Carboxymethyl Rice Starch Films: Sodium Hydroxide Effect"

_molecules, 2022, doi:10.3390/molecules27020331_

Round 1

Reviewer 1 Report

CMSr is a very interesting material that is broadly studied as potential packaging material. The present work analyses the effect of NaOH concentration on the synthesis of CMSr and the quality of the produced films, obtained by casting in the presence and absence of sorbitol as the plasticizer. However, the manuscript contains several issues that authors must improve and clarify to meet the quality level of the Journal. Results need to be explored in deep. XRD information should be provided to support several experimental results. Consequently, the manuscript is not acceptable in the present form. My recommendation: major revisions

  1. My comments 

Please check English grammar and spelling. 

  1. Abstract. This section needs further work.

Line 38 “…. became more of an agglomeration….”. rephrase this sentence

Line 38 “… the mechanical properties including the percentage of solubility….” Solubility and WVP are not mechanical properties. Please edit the sentence

Line 42 “… conventional films”. What does it mean?

Line 40. “The TS value of CMSr film was 40 higher than that of CMSr films with sorbitol as the plasticizer when compared at the same concentration”. Concentration of what? CMSr? Sorbitol?. The abstract should give accurate information about the work. Please clarify

Line 43. “While the %E of CMSr films with sorbitol as the plasticizer was higher than CMSr film, the CMSr films with sorbitol as the plasticizer had better film elongation ability.” This sentence is redundant. 

  1. Introduction

 Introduction should offer a detailed account of the subject under study. Therefore, a description of the literature available on active CMSr synthesis using NaOH and the obtained films should be provided. The first paragraph should be rewritten focusing on the subject of the paper which is CMsr synthesis and resultant films. The effect of NaOH on CMSr synthesis should be introduced. The influence of crosslinking agents, plasticizers, and so on, on the physicochemical properties of CMSr films should be reviewed (see. Li et al., 2011, http://dx.doi.org/10.4028/www.scientific.net/AMR.233-235.306; Rachtanapun et al., 2012, actual ref. 19). Mechanical and dynamic mechanical analysis, Wilpiszewska et al. 2020, Polymers 2020, 12, 2447; doi:10.3390/polym12112447). 

  1. Results and discussion

Describe the reaction of the synthesis of CMSr

Line 88. Authors said: “Swelling of powder with extended surface space for the etherification transform to produce starch aloxides”. I am sorry but I do not understand this sentence.

Line 91. However, (insert WHEN) the NaOH concentration was 50%, the DS of CMSr was lowered due to the degradation of starch.

Line 92. The next sentence, starting: “Furthermore, 92 Rachtanapun et al. [22]”… lacks a verb.

I suggest comparing the obtained results with those informed by other authors on the same reaction performed on starch, instead of cellulose. Differences between both polymers should be discussed. 

Line 112. Again.. “ became more of an agglomeration…” Please rewrite this sentence.

Line 118. Authors stated: Alkalization changed the granules making them weaker with a loss of 117 crystallinity.” Could you support this statement? SEM observations must be supported by XRD diffractograms. 

Line 98. Replace concentration by concentration

TSM of the films. Authors must relate the morphology of the starch granules with TSM of the films. If granules are poorly damaged, how does this influences TSM of the produced films? Do you still have the granules in the films? How does plasticization work?

Lines 144-164 and Fig. 2. Moisture absorption results little contributes to the research. It is well known that the hydrophilicity of hydroxymethylated polysaccharides increases with DS level.

Line 169. Please clarify: you compare films with and without sorbitol?This part is rather confusing. Authors talk about granules, and also about gel….please clarify

If films were obtained after gelatinization, no granules remain. Therefore, tensile properties should be analyzed at the light of the interactions in unplasticized and plasticized CMSr films. Line 180. There you explained this. 

Line 184. E slightly increased with NaOH.” For which film? 

Line 188. Crystallinity changes. Please give experimental evidence. 

Lines 188-189. Could you provide the moisture content (MC%) of the films analyzed? 

Line 219. WVP was reduced with NaOH (fig. 6). This statement contradicts Fig. 6, where WVP increases from 0.27 up to 2.22 (please check the units) for 10 and 60 % NaOH, respectively. If changes in WVP are attributed to changes in crystallinity, this property must be studied.  

Author Response

Review 1

English language and style

( ) Extensive editing of English language and style required
(x) Moderate English changes required
( ) English language and style are fine/minor spell check required
( ) I don't feel qualified to judge about the English language and style

Yes

Can be improved

Must be improved

Not applicable

Does the introduction provide sufficient background and include all relevant references?

( )

( )

(x)

( )

Is the research design appropriate?

( )

( )

(x)

( )

Are the methods adequately described?

( )

(x)

( )

( )

Are the results clearly presented?

( )

( )

(x)

( )

Are the conclusions supported by the results?

( )

(x)

( )

( )

Comments and Suggestions for Authors

CMSr is a very interesting material that is broadly studied as potential packaging material. The present work analyses the effect of NaOH concentration on the synthesis of CMSr and the quality of the produced films, obtained by casting in the presence and absence of sorbitol as the plasticizer. However, the manuscript contains several issues that authors must improve and clarify to meet the quality level of the Journal. Results need to be explored in deep. XRD information should be provided to support several experimental results. Consequently, the manuscript is not acceptable in the present form. My recommendation: major revisions

Thank you very much for helping us improve the revised version of the manuscript. As you suggested and commented, we have addressed all questions and performed the additional experiments. We hope that these changes make you satisfactory.

  1. My comments 

Please check English grammar and spelling.

[Answer]: We have collected the grammar and spelling as you commented.

  1. Abstract. This section needs further work.

[Answer]: We have added the future work in such a section. “However, the tensile strength of the CMSr films was relatively low. Therefore, such a property need to be improved and the application of the developed films should be investigated in the future work”.  

Line 38 “…. became more of an agglomeration….”. rephrase this sentence

[Answer]: We have deleted such a sentence and revised the abstract as you commented.

Line 38 “… the mechanical properties including the percentage of solubility….” Solubility and WVP are not mechanical properties. Please edit the sentence

[Answer]: We have revised the abstract as you commented.

Line 42 “… conventional films”. What does it mean?

[Answer]: We have deleted “conventional film” in the revised version of manuscript.

Line 40. “The TS value of CMSr film was 40 higher than that of CMSr films with sorbitol as the plasticizer when compared at the same concentration”. Concentration of what? CMSr? Sorbitol?. The abstract should give accurate information about the work. Please clarify

[Answer]: We have removed a series of the CMSr films without the sorbitol because this work is only focus on the effect of different NaOH concentrations on the properties of CMSr films.

Line 43. “While the %E of CMSr films with sorbitol as the plasticizer was higher than CMSr film, the CMSr films with sorbitol as the plasticizer had better film elongation ability.” This sentence is redundant. 

[Answer]: We have deleted such sentence as you commented.

  1. Introduction

Introduction should offer a detailed account of the subject under study. Therefore, a description of the literature available on active CMSr synthesis using NaOH and the obtained films should be provided. The first paragraph should be rewritten focusing on the subject of the paper which is CMsr synthesis and resultant films. The effect of NaOH on CMSr synthesis should be introduced. The influence of crosslinking agents, plasticizers, and so on, on the physicochemical properties of CMSr films should be reviewed (see. Li et al., 2011, http://dx.doi.org/10.4028/www.scientific.net/AMR.233-235.306; Rachtanapun et al., 2012, actual ref. 19). Mechanical and dynamic mechanical analysis, Wilpiszewska et al. 2020, Polymers 2020, 12, 2447; doi:10.3390/polym12112447). 

[Answer]: As you commented, we have provided a detailed account of the subject under study in the revised version of the manuscript as the yellow highlighted text.

  1. Results and discussion

Describe the reaction of the synthesis of CMSr

Line 88. Authors said: “Swelling of powder with extended surface space for the etherification transform to produce starch aloxides”. I am sorry but I do not understand this sentence.

[Answer]: As you commented, we have rewritten such sentence in the revised version of the manuscript.

Line 91. However, (insert WHEN) the NaOH concentration was 50%, the DS of CMSr was lowered due to the degradation of starch.

[Answer]: The word “when” was inserted. “However, when the 50% NaOH concentration, the DS of CMSr powder was gradually decreased.”

Line 92. The next sentence, starting: “Furthermore, 92 Rachtanapun et al. [22]”… lacks a verb.

[Answer]: We have edited the content so this sentence “Furthermore, 92 Rachtanapun et al. [22]” was removed.

I suggest comparing the obtained results with those informed by other authors on the same reaction performed on starch, instead of cellulose. Differences between both polymers should be discussed. 

[Answer]: As you suggested, we have revised it as followed.

“There are also previous researches relating to carboxymethyl cellulose that determined the effect of NaOH concentrations on the degree of substitution (DS). For example, carbox-ymethylcellulose from asparagus stalk end (CMCas) with various NaOH concentrations by Klunklin et al. [27]. The results showed that CMCas at a concentration of 30% of NaOH for the carboxymethylation reaction which has a maximum DS of 0.98. In addition, Rachtanapun et al. [28] studied carboxymethyl cellulose from nata de coco (CMCn) with Acetobacter xylinum starting NaOH concentrations from 20% to 60%. In the carboxymeth-ylation process, optimal conditions include using a NaOH content of 30 g/100 mL as the highest DS value (0.92). Including the experiment of Rachtanapun et al. [34]¸ carboxyme-thyl durian rind (CMCd) was synthesized using different concentrations of NaOH. The results showed that the DS value of CMCd increased with increasing NaOH concentration and obtained a maximum DS of 0.87 at a 30% (w/v) NaOH concentration”.

Line 112. Again.. “ became more of an agglomeration…” Please rewrite this sentence.

[Answer]: As you commented, we have rewritten such sentence in the revised version of manuscript. “At 50% NaOH concentration, the CMSr powder exhibited an agglomerated form (Figure 1f-1F) whereas the CMSr treated with 60% NaOH showed a gel-like aspect in which isolated granules of CMSr was obviously disappeared (1g-1G).”

Line 118. Authors stated: Alkalization changed the granules making them weaker with a loss of 117 crystallinity.” Could you support this statement? SEM observations must be supported by XRD diffractograms. 

[Answer]:  As you suggested, we have rewritten the SEM, XRD, and DSC results in the revised version of the manuscript. As described, the XRD and DSC results indicated the reduction and loss of crystallinity of the rice starch, which attributed to the weaken of intermolecular force (H-bond) between the native rice starch molecules and great substitution of carboxymethyl group affected by the higher NaOH concentrations. These results were consistent with the SEM result of the native rice starch and the CMSr powders.

Line 98. Replace concentration by concentration

[Answer]:  We are sorry. We don’t understand what is Replace concentration by concentration.

TSM of the films. Authors must relate the morphology of the starch granules with TSM of the films. If granules are poorly damaged, how does this influences TSM of the produced films? Do you still have the granules in the films? How does plasticization work?

[Answer]: The percentage of soluble matter (%SM) could be changed depending on the morphology of the samples and DS, which affected the dynamic water contact angle of the CMSr films.  As the NaOH concentrations increased, the crystallinity decreased as confirmed by XRD and DSC results.

How does plasticization work?

[Answer]: We have removed a series of the CMSr films without the sorbitol because this work is only focus on the effect of different NaOH concentrations on the properties of CMSr films.

Lines 144-164 and Fig. 3. Moisture absorption results little contributes to the research. It is well known that the hydrophilicity of hydroxymethylated polysaccharides increases with DS level.

[Answer]:  As you commented, we have removed the results of moisture absorption in the revised version of manuscript.

Line 169. Please clarify: you compare films with and without sorbitol? This part is rather confusing. Authors talk about granules, and also about gel….please clarify

[Answer]:  In this study, we mainly focus on the effect of different NaOH concentrations on the properties of the CMSr films. In addition, to prevent confusion, we have remove the tensile properties of the CMSr films without sorbitol in the revised version of the manuscript.

If films were obtained after gelatinization, no granules remain. Therefore, tensile properties should be analyzed at the light of the interactions in unplasticized and plasticized CMSr films. Line 180. There you explained this. 

[Answer]: We have removed a series of the CMSr films without the sorbitol because this work is only focus on the effect of different NaOH concentrations on the properties of CMSr films.  In this research, plasticizer was fixed at 30% in all CMSr films. Therefore, the effect of plasticizer content was not studied. However, in general, plasticizer will not form a covalent bond, it will only disperse in the polymer matrix to make polymer softer and more flexible as well as increase its plasticity.

Line 184. E slightly increased with NaOH.” For which film? 

[Answer]: We have removed the tensile properties of the CMSr film without sorbitol and rewritten this part in the revised version of the manuscript.  Therefore, such film means the CMSr films (with sorbitol). “The %E of the CMSr films slightly increased with increasing NaOH concentrations, indicating the behavior of flexible CMSr films”.

Line 188. Crystallinity changes. Please give experimental evidence. 

[Answer]:  As you commented, we have rewritten and described clearly relating to changes in crystallinity or reduction of crystallinity in the XRD and DSC results, in the revised version of manuscript. The XRD and DSC analysis evidently indicated a reduction of crystallinity as shown in the change in peak intensity and XRD patterns as well as as reflected in the change in Tm and melting enthaphy affected by the higher NaOH concentrations.

Lines 188-189. Could you provide the moisture content (MC%) of the films analyzed? 

[Answer]:  We have analyzed the moisture content and the data were added in the section 3.4. “The CMSr films with 12-14% of moisture content were kept at 27 ± 2 °C and 54 ± 2% relative humidity (RH) for 24 h [25].”

Line 219. WVP was reduced with NaOH (fig. 6). This statement contradicts Fig. 6, where WVP increases from 0.27 up to 2.22 (please check the units) for 10 and 60 % NaOH, respectively. If changes in WVP are attributed to changes in crystallinity, this property must be studied.

[Answer]: As you commented, we have checked the unit and found that it was mistaken. Therefore, we have changed the unit of the WVP, from “mm.g/day-1/m2/kPa-1” to “g.m2 × 10-4 .m-2 .day-1.mmHg”. In addition, we have performed the dynamic water contact angle, XRD, and DSC to confirm that the change in WVP of rice starch film and CMSr films were attributed to changes in chemical affinity of polymeric film to water molecules and change in crystallinity. We have clearly described such properties in the revised version of the manuscript.

Reviewer 2 Report

The manuscript is focused on preparation of carboxymethyl rice starch film by high content of NaOH. The writing is good and the result is interesting. However, some questions should be dealt with:

  1. Could authors show the figures of film product? SEM, AFM and other images were also needed to show the molecular structure of this film.
  2. When NaOH concentration increased from 20% to 30%, SM was dramatically increased while other indices were not, why?
  3. The mechanism of adding sorbitol for the change of film properties?
  4. The great increase of WVP of 50% NaOH, what is the relationship between this and film structure?

Please add characteristics of CMS films like FTIR, XRD, etc., for better understanding of their structures.

Author Response

Review 2

English language and style

( ) Extensive editing of English language and style required
( ) Moderate English changes required
(x) English language and style are fine/minor spell check required
( ) I don't feel qualified to judge about the English language and style

Yes

Can be improved

Must be improved

Not applicable

Does the introduction provide sufficient background and include all relevant references?

(x)

( )

( )

( )

Is the research design appropriate?

( )

(x)

( )

( )

Are the methods adequately described?

(x)

( )

( )

( )

Are the results clearly presented?

( )

(x)

( )

( )

Are the conclusions supported by the results?

(x)

( )

( )

( )

Comments and Suggestions for Authors

The manuscript is focused on preparation of carboxymethyl rice starch film by high content of NaOH. The writing is good and the result is interesting. However, some questions should be dealt with:

Thank you very much for helping us improve the quality of this work. As you suggested and commented, we have addressed all questions and performed the additional experiments including FT-IR, SEM. XRD, DSC, as well as dynamic water contact angle. We hope that these changes make you satisfactory.

  1. Could authors show the figures of film product? SEM, AFM and other images were also needed to show the molecular structure of this film.

[Answer]: As you suggested, we have performed the additional experiments (i.e., FT-IR, SEM, XRD, DSC, as well as dynamic water contact angle) to improve the quality of this work. However, we could not access the AFM. In the revised version of manuscript, we have additionally described the changes in molecular structure, morphology, crystallinity, and hydrophilicity of the films affected by different NaOH concentrations during carboxymethylization reaction. As described, the FT-IR results clearly indicated the the appearance of carboxymethyl group on the surface of CMSr films treated with higher NaOH concentrations confirming the a greater substiution of carboxymethyl group. In addition, XRD and DSC indicated the decrease in peak intensity and a reduction of melting enthaphy, implying a reduction of crystallinity. These changes resulted in the increase in hydrophilicity of the CMSr films as indicated in the reduction of dynamic water contrat angles.

  1. When NaOH concentration increased from 20% to 30%, SM was dramatically increased while other indices were not, why?

[Answer]: As you commented, we have rewritten the %SM part. At lower NaOH concentrations (10-20%), the %SM of the CMSr films was low due to insufficient NaOH concentrations for carboxymethylation reaction. Conversely, at higher NaOH concentrations (30-50%), the %SM of the CMSr films was obviously high because the higher NaOH concentrations were sufficient to react with the rice starch molecules, resulting in the breaking of H-bond (in rice starch granules) and reduction of crystallinity. This led to the increase in carboxymethylation reaction of CMSr powders. As consequently, %SM of CMSr films treated with higher NaOH concentrations dramatically increased. This result was consistent with DS, FT-IR, and SEM results.

  1. The mechanism of adding sorbitol for the change of film properties?

[Answer]: We have removed a series of the CMSr films without the sorbitol because this work is only focus on the effect of different NaOH concentrations on the properties of CMSr films.

  1. The great increase of WVP of 50% NaOH, what is the relationship between this and film structure?

[Answer]: The CMSr film treated with 50% NaOH exhibited the great WVP due to the higher DS and lower dynamic water contact angle (a higher chemical affinity between film and water molecules) as well as a lower crystallinity, leading to the great WVP.

Please add characteristics of CMS films like FTIR, XRD, etc., for better understanding of their structures.

[Answer]: As you suggested, we have performed the additional experiments including FT-IR, SEM, XRD, DSC, as well as dynamic water contact angle to improve the quality of the revised version of the manuscript.

Reviewer 3 Report

The manuscript investigated the influence of NaOH on the morphological, mechanical, and water barrier of 2 Carboxymethyl Rice Starch Films. The manuscript is well written and organized. However, there are some issues that need to be clear before publishing. Therefore, I would suggest accepting the manuscript with the Major revision. 
I mentioned some points to improve the manuscript:
1.    The authors should control the text's grammatical, typos errors, and references.
There are some examples:
•    In segment 2.1., Please add space between numbers and units, check all.
•    Please add space between the end of the sentences (check all). 
2.    In the introduction:
•    Line 53: “However,”, instead of “Although”
•    Line 58: The most generally “plasticizers”
3.    In the Results and Discussion:
•    Please add the chemical schematic of the effect of NaOH on the CMS.
•    Please perform ATR-FTIR and Thermal analysis to confirm the result. 
•    In figure 2, remove 60% since nothing report.
4.    In Materials and Methods:
•    Please correct the format of equations 4 and 5
•    Correct the number of equations after 4, both in front of equations and in the text.

Author Response

Review 3

English language and style

( ) Extensive editing of English language and style required
( ) Moderate English changes required
(x) English language and style are fine/minor spell check required
( ) I don't feel qualified to judge about the English language and style

Yes

Can be improved

Must be improved

Not applicable

Does the introduction provide sufficient background and include all relevant references?

( )

( )

(x)

( )

Is the research design appropriate?

( )

(x)

( )

( )

Are the methods adequately described?

( )

( )

(x)

( )

Are the results clearly presented?

( )

(x)

( )

( )

Are the conclusions supported by the results?

( )

( )

(x)

( )

Comments and Suggestions for Authors

The manuscript investigated the influence of NaOH on the morphological, mechanical, and water barrier of 2 Carboxymethyl Rice Starch Films. The manuscript is well written and organized. However, there are some issues that need to be clear before publishing. Therefore, I would suggest accepting the manuscript with the Major revision. 
I mentioned some points to improve the manuscript:
1.    The authors should control the text's grammatical, typos errors, and references.
There are some examples:
•    In segment 2.1., Please add space between numbers and units, check all.
•    Please add space between the end of the sentences (check all). 

Thank you very much for helping us improve the revised version of the manuscript. As you suggested and commented, we have addressed all questions and performed the additional experiments. We hope that these changes make you satisfactory.

[Answer]: As you suggested, we have rewritten and cleaned up the text's grammatical, typos errors, and references.

  1.    In the introduction:
    •    Line 53: “However,”, instead of “Although”

[Answer]: We replaced the word “However” by “Although”, as you commented.

  •    Line 58: The most generally “plasticizers”

[Answer]: We replaced the word “plastic” by “plasticizers”.

  1.    In the Results and Discussion:
    •    Please add the chemical schematic of the effect of NaOH on the CMS.

[Answer]: As you suggested, we have added the chemical schematic of the effect of NaOH on the CMS.

  •    Please perform ATR-FTIR and Thermal analysis to confirm the result. 

[Answer]: As you suggested, we have performed the ATR-FTIR and thermal analysis to confirm the results. 

  •    In figure 2, remove 60% since nothing report.

[Answer]: As you suggested, we have removed the 60% NaOH in the revised version of the manuscript.

  1.    In Materials and Methods:
    •    Please correct the format of equations 4 and 5
    •    Correct the number of equations after 4, both in front of equations and in the text.

[Answer]: As you commented, we have corrected the format and number of equations in the revised version of the manuscript.

Round 2

Reviewer 1 Report

Revised manuscript is acceptable in the present form

Reviewer 3 Report

The manuscript has been improved, and it is eligible for publishing in this journal.